# Understanding pathways to stimulant use: a mixed-methods examination of the individual, social and cultural factors shaping illicit stimulant use across Europe (ATTUNE): study protocol

Moritz Rosenkranz,[1] Amy O'Donnell,[2] Uwe Verthein,[1] Heike Zurhold,[1] Michelle Addison,[3] Nienke Liebregts,[4] Magdalena Rowicka,[5] Miroslav Barták,[6] Benjamin Petruželka,[6] Eileen FS Kaner,[2] Marcus-Sebastian Martens[1]

For numbered affiliations see end of article.

**Correspondence to**
Moritz Rosenkranz;
moritz.rosenkranz@uni-hamburg.de

## ABSTRACT

**Introduction** Amphetamine-type stimulants (ATS) including amphetamine, methylenedioxymethamphetamine/'ecstasy', methamphetamine, synthetic cathinones and 'Ritalin' are the second most commonly used illicit drugs globally. Yet, there is little evidence on which factors are associated with the development of different patterns of ATS use over the life course. This study aims to examine which individual, social and environmental factors shape different pathways and trajectories of ATS consumption. The study will be conducted in five European countries: Germany, the Netherlands, Poland, Czech Republic and the UK.

**Methods and analysis** We will use a sequential mixed-methods study design to investigate the multiple factors (familial, social and occupational situation, critical life events, general risk behaviour, mental and physical health, satisfaction with life) that shape individual ATS use pathways. A systematic literature review will be performed to provide an overview of the current academic literature on the topic. In module 1, qualitative semistructured interviews (n=ATS users and non-users) will be conducted to explore individual experiences of, and perspectives on, dynamics of change in stimulant consumption patterns. In module 2, structured questionnaires (n=2000 ATS users and non-users) will be administered via tablet computers to validate and enhance the generalisability of the interview findings. Data integration will take place at two key points. First, during the study, where the findings from the first qualitative interviews will inform the design of the structured questionnaire. Second, at the end of the study, where mixed methods data will be brought together to generate an in-depth, contextualised understanding of the research topic.

**Ethics and dissemination** The study has been approved by the respective responsible ethics committee in each participating country. Data will be treated confidentially to ensure participants' anonymity. Findings will be disseminated in peer-reviewed scientific journals, national and international conferences, and in briefings for policy and practice.

### Strengths and limitations of this study

► First study of its kind to examine different pathways of amphetamine-type stimulant (ATS) use and will thus expand an underdeveloped evidence base.
► Large qualitative and quantitative dataset collected in five European countries, which allows valuable intercountry comparisons.
► Applies a theory-based analytical framework to understand the individual, social and environmental influences shaping ATS use, which will provide important insights expedient for the development of tailored prevention and intervention measures.
► Cross-sectional design allows correlative but no causal conclusions.

## INTRODUCTION

Amphetamine-type stimulants (ATS) are the second most commonly used illicit drugs globally[1] as well as in Europe.[2] ATS include amphetamines ('speed'), methylenedioxy-methamphetamine ('MDMA' or 'ecstasy'), methamphetamine ('crystal meth')[3] and illicit use of amphetamine-type prescription drugs (eg, 'Ritalin'). In recent years, there has also been a rise in new psychoactive substances that mimic the effects of stimulants in global drug markets,[4] including synthetic cathinones such as mephedrone ('bath salts'). Across Europe as a whole, 0.5% of those aged 15–64 reported using amphetamines in the past 12 months, with higher rates for MDMA use (0.8%). However, ATS use rates vary by country, with the highest consumption found in the Netherlands (amphetamine, 1.7%; MDMA, 3.6%). Data also suggest higher consumption rates for adolescents and younger adults compared

with the general population, with 1% of those aged 15–34 reporting amphetamine consumption and 1.8% MDMA consumption, over the previous year.[1 2] Worldwide, quantities of ATS seized have doubled over the past decade,[5] with increased usage levels reflected in wastewater analyses conducted in several European countries[6] and in a corresponding rise in numbers of first-time entrants for stimulant treatment across Europe from about 7000 in 2006 to more than 12 000 in 2016.[2] Long-term use of ATS can lead to a substance use disorder, including (psychological) dependency.[7] The global prevalence of substance use disorders related to amphetamines was estimated at nearly five million people in 2016. In Europe around 260 000 people are affected by amphetamine use disorder, with prevalence twice as high in Eastern Europe as in the central region.[8]

Methamphetamine use at dependent levels is associated with multiple comorbidities, including HIV infection, hepatitis, cardiac effects, cognitive dysfunction and prominent psychiatric consequences such as psychosis.[9 10] Although MDMA is often viewed as a recreational drug, prolonged use is associated with neurological dysfunction and depression.[11] Additional societal costs identified with ATS abuse include premature death, crime, lost productivity, environmental damage, disruption of family life and infectious disease.[9 12 13] However, despite the substantial harms associated with ATS use, increased prevalence in consumption rates and rising number of treatment entries, there is little evidence regarding which factors shape different patterns of ATS use.

Some qualitative and very few quantitative studies exploring influences on ATS use have been published, primarily focused on factors affecting initiation.[14] Important motives for the initiation of ATS consumption identified in previous research include: curiosity or propensity for experimentation[15–18]; self-management of stress, trauma or other mental health issues[16 18–24] and to boost of performance at work/studies[19 21 25–27] or in private settings (sexual relationships, endurance at dance events).[18 26 28–30] There is some evidence to suggest that continued and/or increased ATS consumption is often to support specific functional needs (improvement of stress management or reduced insecurity in social situations) and to help manage withdrawal effects.[15 20 22 27 29 31–34] Experiencing critical life events (separation, death of a close friend or family member, domestic violence) also appears to be associated with sustained use.[19 26 32] Reported factors connected with a decrease (and in some cases, until abstinence) included an increased perception of negative health impacts,[16 20 29 35 36] changing social networks and reduced availability of ATS.[28 37 38] However, there remains limited understanding of what influences different trajectories of consumption over time and whether this varies by type of ATS substance or user characteristics, such as gender, age or socioeconomic status.

In the framework of the European Research Area Network on Illicit Drugs, a consortium of five research institutions from Germany, the UK, Poland, the Netherlands and the Czech Republic, was formed to conduct a study to respond to this evidence gap. The European ATTUNE study (Understanding Pathways to Stimulant Use: a mixed-methods examination of the individual, social and cultural factors shaping illicit stimulant use across Europe) is led by Germany, as the principal investigator.

## Research objectives

The overall aim of the project is to improve our understanding of which factors shape different pathways of ATS use in Europe. By examining interactions between individual, social and environmental influencing factors, and the overall trajectory of drug use, this study will explore individual motivations to use ATS and describe different patterns of consumption over time. In doing so, the study seeks to identify potential protective factors (eg, personality traits, social integration) associated with the ability to control, decrease or quit ATS use, as well as risk factors (eg, critical life events) associated with the escalation of ATS consumption patterns towards problematic use and/or dependence. Further, we aim to explore why some individuals exposed to ATS select not to use these substances, as well as examining the relationship that illicit stimulant users have with other illicit and licit substances. Targeted recruitment of different ATS user groups with regard to frequency, dependency, former or current use will ensure a sample that varies by type and level of ATS use. Details about the different user groups targeted by this study are provided below.

## METHODS AND ANALYSIS
### Study design

ATTUNE is a sequential, exploratory, mixed-methods study. The design, implementation and interpretation of the study is informed by the biopsychosocial model of substance use.[39] This model suggests that the change process of drug use pathways is influenced by the interaction of three core domains: individual differences, social dynamics and the environmental/cultural setting. The study comprises of three main components.

First, a systematic review of the qualitative and quantitative published literature on which individual, social and environmental influences shape different pathways of ATS use over the life course. The qualitative literature review is completed and published.[14] We searched four databases for peer-reviewed qualitative studies which explored the views of ATS users on which factors have shaped their drug use careers. The search strategy was conducted in accordance with the Sample, Phenomenon of Interest, Design, Evaluation, Research type (SPIDER) tool.[40] Further details can be found in the corresponding publication. The ongoing systematic review of the quantitative literature (studies on ATS use in adults and adolescents) aims to appraise the evidence on risk, protection, resilience and desistance. Search strategy as well as data extraction are analogous to the qualitative review. The

findings from these reviews will provide an insight into the existing international literature that examines ATS users' perspectives on why they start, stop, increase and/or reduce their ATS consumption while also exploring under which circumstances and conditions stimulant users change their consumption patterns.

Next, in module 1 of the fieldwork, qualitative methods (semistructured interviews) will be used to explore individual experiences and perspectives on dynamics of change in stimulant consumption patterns. The topic guide will build in particular on the findings from the review of qualitative literature conducted at the start of the project, as well as the theoretical underpinning for the research (biopsychosocial model of drug use).

Finally, in module 2, the findings from the qualitative interviews will inform the development of a structured questionnaire to validate and enhance the generalisability of the results in a large sample of ATS users and non-users.

The total length of the study is 36 months (September 2016 to August 2019). The estimated duration of the recruitment as well as data collection for module 1 is 7 months, and for module 2 it will amount to 13 months.

## Participants
### Eligibility criteria
Individuals who have either used or had the opportunity to use ATS are eligible for inclusion in both the qualitative interviews and survey questionnaire. This includes people who have either consumed ATS at least once in their life or people who have never used ATS but have been exposed to ATS consumption (defined as having been present when family or friends took ATS but refused to consume themselves). To ensure the inclusion of only those who have had the opportunity to experience changes in the trajectory of their ATS use, the participant's first ATS consumption (or exposure) needs to have taken place at least 5 years before the interview or survey questionnaire. We also excluded people previously diagnosed with opioid dependence (self-reported) to avoid overlap between pathways to opioid use and pathways to ATS use. Excluding these participants also ensures that our sample is not dominated by former or current opioid users who consume stimulants primarily to complement their opioid use (eg, to get a 'kick' while in opioid substitution therapy).

Further inclusion criteria
► Aged 18 years or older.
► Resident in one of the five national sampling regions.
► Able to take part in the interview (not psychotic, no severe cognitive impairments or language barriers).

A screening website will be set up where interested persons can check their eligibility to participate in an interview or survey questionnaire. If a person is screened successfully, a message is displayed inviting the person to participate and offering different possibilities to contact the research team to arrange an interview appointment. A randomly chosen screening ID facilitates the connection of the screening data with the interview while maintaining anonymity.

### Study groups
Participants in both modules 1 and 2 will be recruited using convenience sampling (see below). To ensure a sufficient variety of ATS use patterns or ATS use trajectories, six study groups were predefined for module 1 and five study groups for module 2. This approach provides a stratified sample, in which each ATS use pattern (group) serves as a stratum.

*Module 1: Qualitative semistructured interviews*
Eligible participants meeting the above criteria will be allocated to one of six study groups depending on their ATS consumption patterns (time, frequency, dependency). Table 1 shows an overview of the operationalisation of the six groups.

We defined the current use as ATS consumption within the previous 12-month period. Frequent use was defined as those reporting ATS consumption on 10 or more occasions (consumption days) during the previous 12 months (groups 1 and 3) or any 12-month period prior to the past year (groups 2 and 4). ATS dependency was assessed using the Severity of Dependence Scale (SDS).[41] We chose a cut-off of 4 points or more to identify ATS dependency.[42]

*Module 2: Survey questionnaire*
Eligible participants meeting the above criteria will be allocated to one of five study groups depending on their ATS consumption patterns (time, frequency, dependency). table 2 shows an overview of the operationalisation of the five groups. To allow us to distinguish clearly between the current and previous use, we defined the current use as ATS consumption in the past 3 months and the former use as no consumption in the past 12 months. This means that participants reporting ATS use over 3 but less than 12 months previously are excluded.

### Sample size
*Module 1: Qualitative semistructured interviews*
Forty-five persons per study group (n=270 for the total sample, see table 3) were considered sufficient for module 1.[43] As we plan to recruit participants via purposeful sampling,[44] we expect to generate data that are rich enough to answer our research questions and inform the questionnaire used in module 2 of the study.

*Module 2: Survey questionnaire*
Statistical analyses within the group of current users (A_1 and A_2) as well as within the group of former users (B_1 and B_2) are planned (see table 2). To facilitate this at country level, a group size of n=100 is sufficient to detect statistically significant small to medium effects for continuous distributed variables (d≥0.40, α=0.05, power=80%).[45] The sample sizes of the groups are planned as follows. For both, groups A and B, up to 200 participants in each country will be recruited. This allows us to analyse subgroups, for example, dependent and non-dependent

**Table 1** Operationalisation of study groups in module 1

| Study groups in module 1 | Name | Past 12 months prevalence | ≥10 consumption days within past 12 months | ≥10 consumption days within 1 year (at any time except past 12 months) | Currently ATS dependent | Formerly ATS dependent |
|---|---|---|---|---|---|---|
| Group 1 | Currently dependent users | Yes | Yes | n.a. | Yes | n.a. |
| Group 2 | Formerly dependent users | n.a. | n.a. | Yes | No | Yes |
| Group 3 | Currently frequent, non-dependent users | Yes | Yes | n.a. | No | No |
| Group 4 | Formerly frequent, non-dependent users | No | n.a. | Yes | No | No |
| Group 5 | Non-frequent users (currently or formerly) | No | n.a. | No | No | No |
| Group 6 | Exposed non-users | n.a. | n.a. | n.a. | n.a. | n.a. |

Module 2: Survey questionnaire.
ATS, amphetamine-type stimulant.

users in groups A as well as B. As group C consists of non-users only, the analyses will focus on comparison with one of the user groups (A or B), reducing the required size for this group to n=100. The total sample size for module 2 for all countries will amount to 2000 persons (see table 4). Due to funding restraints, the sample size in the Netherlands and the Czech Republic is smaller, which might impede the production of statistically significant country specific intragroup comparisons.

## Recruitment
### Sampling method and recruitment procedure
Participants in both modules will be recruited by non-probability (convenience) sampling, which is an accepted means of accessing participants from 'hard-to-reach' as well as minority populations.[46 47] Preidentified sampling areas in each participating country (see tables 3 and 4) are designed to include participants living in urban as well as rural areas.

Multiple methods and sites will be employed to identify and recruit participants for both modules. Leaflets and posters containing information about the study, a link to the screening website and contact details of the research teams will be printed and distributed in substance use help and treatment facilities, on university black boards, in head shops, bars and nightclubs. Social media and substance use web forums will be used to share and circulate the request for participants. Participants will also be recruited actively at university campuses, drug service facilities and music festivals by directly approaching potential participants. All interested interviewees will be screened for eligibility prior to the interview. If eligible to participate, each respondent will receive a screening code, and the interview can either be conducted straight away or an appointment can be made for a later date. At the end of each interview, participants will be asked to recruit further participants from their social network.

**Table 2** Operationalisation of study groups in module 2

| Study groups in module 2 | Name | Past 12 months prevalence | Past 3 months prevalence | ≥10 consumption days within past 12 months | ≥10 consumption days within 1 year (at any time except past 12 months) |
|---|---|---|---|---|---|
| Group A_1 | Currently frequent users | Yes | Yes | Yes | n.a. |
| Group A_2 | Currently non-frequent users | Yes | Yes | No | n.a. |
| Group B_1 | Formerly frequent users | No | n.a. | n.a. | Yes |
| Group B_2 | Formerly non-frequent users | No | n.a. | n.a. | No |
| Group C | Exposed non-users | n.a. | n.a. | n.a. | n.a. |

**Table 3** Sample sizes module 1 by countries and study groups

| Country | Partner institution | Data collection regions | Sample sizes | | | | | | Total N countries |
| | | | Group 1 | Group 2 | Group 3 | Group 4 | Group 5 | Group 6 | |
| | | | Currently dependent | Formerly dependent | Currently frequent, non-dependent | Formerly frequent, non-dependent | Non-frequent (currently or formerly) | Exposed | |
|---|---|---|---|---|---|---|---|---|---|
| Germany | ZIS | Border region to Czech Republic /metropolitan region of Hamburg | 10 | 10 | 10 | 10 | 10 | 10 | 60 |
| UK | UNEW | Northern England | 10 | 10 | 10 | 10 | 10 | 10 | 60 |
| Poland | APS | Metropolitan region of Warsaw | 10 | 10 | 10 | 10 | 10 | 10 | 60 |
| Netherlands | RG | Amsterdam/the region of Eindhoven | 10 | 10 | 10 | 10 | 10 | 10 | 60 |
| Czech Republic | OGCR | Border region to Germany | 5 | 5 | 5 | 5 | 5 | 5 | 30 |
| Total N groups | | | 45 | 45 | 45 | 45 | 45 | 45 | 270 |

APS, The Academy of Special Education, Warsaw; OGCR, Office of the Government of the Czech Republic; RG, De Regenboog Groep, Amsterdam; UNEW, Institute of Health and Society, Newcastle University; ZIS, Centre of Interdisciplinary Addiction Research of Hamburg University.

This additional snowball sampling approach will be realised by handing over study leaflets, as well as up to three numbered cards containing contact details of the research team. The numbers on the cards will help to track the snowball sampling approach.

### Procedure
#### Fieldwork
##### Module 1
Face-to-face semistructured interviews will be conducted by members of the research team of each participating country. Prior to the interview, all participants will receive an information leaflet containing details about the study, explaining what participation involves: anonymity, confidentiality, the use of data and data protection rules. The participants will then be invited to complete a verbal consent form, should they wish to participate. The interview will be audio recorded and will last approximately 45 to 60 min. On completion of the interview, each participant will receive an incentive (money or vouchers, depending on country). All interviews will be transcribed in full, transferred to appropriate software for analysis, and the audio file deleted.

##### Module 2
The quantitative survey will be conducted with the computer-assisted personal interviewing method. The questionnaire content will first be developed in Microsoft Word. Once finalised, all questions will be programmed using survey software GessQ to enable administration via password-protected tablet computers. The survey instrument will be translated and piloted in all partner countries and revised as necessary. The latest version of the questionnaire will be hosted on a central server operating in the information technology environment of Hamburg University and can be downloaded to the tablets directly as necessary. This server is also the recipient for the data uploads from the tablets.

In each country, trained research assistants will recruit participants and conduct the interviews face-to-face or via video-telephony (Skype). Show cards containing relevant prompts and additional information (eg, lists of ATS, answering scales) will support the conduct of the interviews.

### Monitoring and data management
The screening process and fieldwork progress will be monitored using a study-specific coordination database. This database will be populated with screening data from each participant as well as key information regarding the interviews or questionnaires, such as study group, duration of interview, gender distribution, sampling region and contact with drug help services.

#### Module 1
The raw qualitative data from module 1 will be managed, stored and analysed by the respective research teams in each participating country. Initial analyses will be conducted at country level using a common coding framework. Analysed data, including emerging interview themes, will be pooled and provided to ATTUNE PI (Hamburg University).

#### Module 2
The raw quantitative data will be uploaded continuously to the central data management at the Hamburg University. The raw data will be cleaned, edited and transformed into SPSS datasets, one for each country, as well as one comprehensive dataset covering all five countries.

**Table 4** Sample sizes module 2 by countries and study groups

| Country | Partner institution | Data collection regions | Sample sizes | | | | | Total N countries |
|---|---|---|---|---|---|---|---|---|
| | | | Current ATS user | | Former ATS user | | Non ATS user | |
| | | | A_1: Frequent | A_2: Non-frequent | B_1: Frequent | B_2: Non-frequent | C: exposed | |
| Germany | ZIS | Border Region to Czech Republic / metropolitan region of Hamburg | 100 | 100 | 100 | 100 | 100 | 500 |
| UK | UNEW | Northern England | 100 | 100 | 100 | 100 | 100 | 500 |
| Poland | APS | Metropolitan region of Warsaw | 100 | 100 | 100 | 100 | 100 | 500 |
| Netherlands | RG | Amsterdam/the region of Eindhoven | 50 | 50 | 50 | 50 | 50 | 250 |
| Czech Republic | OGCR | Border region to Germany | 50 | 50 | 50 | 50 | 50 | 250 |
| Total N groups | | | 400 | 400 | 400 | 400 | 400 | 2000 |

## Survey instruments

### Module 1: Interview guideline and time sheet

Two semistructured interview guides will be used to conduct the in-depth interviews in module 1: one for the ATS user groups plus an adapted topic guide for the non-user group. The topic guide will be based on key emergent themes from the systematic literature review as well as relevant theoretical considerations (biopsychosocial model of substance use). Participants will be asked about their experiences and their consumption patterns regarding ATS and other licit and illicit drugs. To obtain a detailed understanding of which influences have shaped these ATS use patterns, participants will be asked about drug use motives, effects/consequences, settings and occasions and how these have changed during the period(s) in life where ATS use took place. The interview will end with questions about the (social) setting of use and its impact and the integration of ATS use into the respondent's lifestyle. The interview guidelines are provided in the online supplementary material. While the use of the interview guide will ensure that all central topics are covered, participants will have the opportunity to discuss additional issues or concerns where relevant. During the interviews, researchers will chart participants substance use over time, including age of onset, frequency and life stage, as well as positive and negative life events (eg, family/partnership, education/work, illness, treatment, imprisonment).

### Module 2: Quantitative questionnaire

The quantitative survey questionnaire will include questions based on the key themes emerging from the qualitative interviews, as well as a selection of standardised instruments to assess various substance use, health and psychological factors (see table 5).

The questionnaire will cover:

a. Sociodemographics: These include sex, age, citizenship, migration background, relationship and children, living situation, educational and occupational situation and social integration.

b. Drug use: Detailed assessment of all illicit drugs ever used in life (lifetime prevalence, past 12-month prevalence, past 30-day prevalence, age at first and last use), test for alcohol dependence (Cutting Down, Annoyance by criticism, Guilty feeling, Eye-Openers (CAGE); Alcohol Use Disorders Identification Test-Consumption (AUDIT-C)) and tobacco smoking status.

c. ATS use: Test for ATS dependence (SDS), injecting drug use and treatment experiences; usual setting of ATS use; patterns, motives and consequences of ATS use including (reasons for) changes.

d. Judicial problems: times and reasons for imprisonment.

e. Physical and mental health assessment (Brief Symptom Inventory-18).

f. Personality assessment (Big Five Inventory-10, Brief Sensation Seeking Scale-4, Generalized Self-Efficacy

**Table 5** Overview of standardised measurement instruments

| Name | Acronym | Content | Reliability: Cronbach's α | Validity: sensitivity/specificity (cut-off) |
|---|---|---|---|---|
| International Standard Classification of Education[52] | ISCED | Identification of highest educational level | — | — |
| Subjective social integration[53] | SSI | Subjective assessment of social integration | — | — |
| Subjective social position[53] | SSP | Subjective assessment of social position | — | — |
| CAGE questionnaire[54] | CAGE | Alcohol problems lifetime | 0.8–0.98 | 0.71/0.90 (2) |
| Alcohol Use Disorders Identification Test[55] | AUDIT-C | Alcohol problems past year | 0.91 | 0.93/0.66 (4) |
| The Severity of Dependence Scale[41] | SDS | ATS dependency lifetime | 0.81–0.89 | 71.3/77.1 (4) |
| Brief Symptom Inventory-18[56] | BSI-18 | Measurement of somatisation, anxiety, depression | 0.87–0.94 | 91.2/92.6 (63) |
| Satisfaction with life scale[57] | SWLS | General life satisfaction | — | — |
| Big Five Inventory[58] | BFI-10 | Assessment of five personality traits | 0.58–0.84 | — |
| Brief Sensation Seeking Scale[59] | BSSS-4 | Measurement of sensation seeking | 0.66 | — |
| Generalised Self-Efficacy Scale[60] | GSE | Measurement of self-efficacy | 0.92 | — |
| Connor-Davidson Resilience Scale[61] | CD-RISC-10 | Measurement of resilience | 0.89 | — |

Scale, Connor-Davidson Resilience Scale-10) and critical life events.

For group C (non-users), some questions will be omitted (eg, questions about ATS use) and replaced with alternative questions focused on their motives for non-use and exposure situation.

### Planned analyses
#### Module 1
Qualitative data from the semistructured interviews will be analysed using content analysis and conducted with appropriate software, such as MAXQDA[48] or NVivo.[49] Partners will develop a common, unified coding system to facilitate comparable findings from the interviews across all partner countries. In addition, the data collected with the timeline and chart on substance use and life events will be merged and analysed systematically. Each partner will conduct initial country-level analysis separately in the respective national language. Once this initial analysis has been completed, each partner will produce a report detailing the country-level findings in English and following a common template. These reports will then be compiled and synthesised to produce a comprehensive, cross-national analysis of the qualitative interviews.

#### Module 2
The quantitative data analysis will be conducted using the statistical software package SPSS V.22.[50] Descriptive, univariate analyses will be used to describe the sociodemographics, health and personality assessment characteristics of the sample, alongside 'consumption careers' (substance use, motives of use and changes of use patterns). Independent sample t-tests and $\chi^2$-analyses, corrected for multiple testing, will be conducted to compare gender and age characteristics among the ATS user groups. Multivariate approaches will be used to assess a wide range of factors derived from the biopsychosocial model regarding their possible influence on ATS use patterns. So, for example, by calculating a multivariate analysis of variance (MANOVA), we can simultaneously test if an individual factor like resilience, a social factor like social integration and an environmental factor like ATS availability is associated with the number of ATS consumption days, the number of cannabis consumption days and the mental health condition index. Furthermore, the MANOVA allows us to detect interactions between the dependent variables. When it comes to an exploration of different ATS user groups such as frequent users, non-frequent users and non-users, we will apply a multinomial logistic regression and determine the association between for example, traits and being a member of one of those groups. If data permit, we will calculate latent class analyses to identify ATS user groups which characteristics are unknown yet. Analyses will be conducted at country level, as well as across the full European sample.

## Patient and public involvement

Patients and members of the public will be involved in ATTUNE at various stages of the study. Policy makers, European non-governmental organisations and service users helped shape the design and focus of the study prior to obtaining funding. We will hold information/discussion sessions about the study with statutory and non-statutory service providers to acquire their insights into how the findings could potentially impact and shape the everyday lives of their service users. Recruited participants enrolled into the study will be invited to act as 'seeds' for the snowball sampling of additional survey interview subjects. This inclusion of patients/public in this way helps with enhanced recruitment and enables these participants to share their experiences of taking part with others and to underline the importance of the study to people like themselves. ATS user representatives and public representatives will be actively involved in disseminating the results of the research.

## Ethics and dissemination

Based on the regulations in each participating country, ethical approval was obtained. Participant anonymity will be maintained in both the semistructured interviews and the survey questionnaire. During the survey, no information will be collected that could link the data to the participant concerns. All participants will be explicitly asked to provide informed consent to taking part in the study and made aware of the data protection rules. A written informed consent form, signed by the project leader, will be made available to each participant before the interview. Verbal consent in module 1 will then be asked for, recorded and documented in the transcription of the interview. The quantitative interview (module 2) starts with the question, if the participant has read the informed consent form and if she/he is willing to give consent. The information about each participant's consent will be saved in a dedicated variable of the dataset.

Research findings of ATTUNE will be disseminated in peer-reviewed, open-access journals as well as at national and international conferences and workshops. Each partner will also deliver reports to their funding institutions under the specific terms of the respective country. Additionally, an accessible report will be drafted and distributed to organisations who express an interest in the study. A specific report that is accessible to substance users will be developed for individuals. We intend to disseminate these findings through social media to maximise impact and expand networks of interest.

## Strengths and limitations of this study

By using multiple methods (systematic literature reviews, qualitative interviews and survey questionnaires), this study will generate in-depth, contextualised evidence in an underexplored field of research. The use of stratified sampling will ensure a sufficient variety of types of ATS users and non-users are included, reflecting different use patterns, current and ex-users, dependent and non-dependent users as well as exposed non-users. In particular, the views and experiences of non-users are rarely reflected in the evidence base and could help generate novel insights into which factors shape decisions not to consume ATS. The application of standardised interview guides and questionnaires will result in a large comprehensive sample that will allow us to compare ATS use in multiple and varied sociocultural, political and legal environments across Europe.

A key limitation of the study is the cross-sectional design, which makes it difficult to trace pathways and trajectories of ATS use over time. We address this by only including persons, whose first contact with ATS consumption occurred at least 5 years ago, meaning they have had the chance to develop different ATS use patterns. However, it is important to stress that a cross-sectional design allows correlative but no causal conclusions. One further issue concerns the convenience sampling approach employed in this study in predefined sampling regions. At the same time, compared with representative general population surveys, our method should ensure increased levels of inclusion of different ATS user types, allowing for more detailed and in-depth insights into different use trajectories. Finally, by defining multiple inclusion criteria as well as precise group strata, we aim to mitigate potential bias that could emerge from this sampling approach.

## IMPLICATIONS FOR INTERVENTIONS AND FUTURE POLICY

The findings from this research will enable policy makers and practitioners to improve existing ATS prevention and intervention programmes and support the development of new approaches in the future. By examining different types of stimulant users (including ex-users and non-users), information will be generated which will be important for universal prevention (targeting general populations), selective prevention (focussing on vulnerable groups) and indicated prevention (aiming at groups that show early signs of problematic substance use). Furthermore, given the limited long-term efficacy of ATS treatment,[51] the findings of this study could support the development of ATS treatment programmes that are more effectively tailored to the needs of specific ATS populations and individual users.

**Author affiliations**
[1]Centre of Interdisciplinary Addiction Research of Hamburg University, Department of Psychiatry, University Medical Centre Hamburg-Eppendorf, Hamburg, Germany
[2]Institute of Health and Society, Newcastle University, Newcastle upon Tyne, UK
[3]Department of Social Sciences, Northumbria University, Newcastle upon Tyne, UK
[4]Bonger Institute of Criminology, University of Amsterdam, Amsterdam, The Netherlands
[5]Maria Grzegorzewska Academy of Special Education, Institute of Applied Psychology, Warsaw, Poland
[6]Department of Addictology, First Faculty of Medicine, Charles University and General University Hospital in Prague, Prague, Czech Republic

**Acknowledgements** We thank all policy makers, European non-governmental organisations and service users that helped shape the design and focus of the study.

**Contributors** MR, MSM, HZ, UV, AOD and MA planned and designed the study. All other authors contributed to the design of the study. UV is the principal investigator; MSM and MR are the overall project managers and coordinators of the study. Country-specific study coordination in the UK is performed by AOD, MA and EFSK in the Netherlands by NL in Czech Republic by MB and BP in Poland by MRow, and in Germany by MR, MSM, HZ and UV. MR wrote the first draft of the study protocol and led the revisions of the manuscript with substantial critical input from all coauthors. In addition to crucial input as regards content, AOD provided significant help to improve the article′s language. All authors read and approved the final version of the manuscript.

**Funding** The superordinate research framework ERANID is funded by the European Union under the 7th Framework Programme. Each project partner receives funding from its national funding bodies: UK: National Institute for Health Research (NIHR) Policy Research Programme (project ref. PR-ST-0416-10001); Germany: Bundesministerium für Gesundheit (project ref. ZMVI1-2516DSM222); Czech Republic: Office of the Government of the Czech Republic (Decision No. 10701635 / 18- OPK) and Charles University (No. PROGRES Q06/LF1); Poland: the National Bureau for Drug Prevention; the Netherlands: ZonMw.

**Competing interests** UV received a speaker's honoraria and travelling expenses from Mundipharma GmbH. AOD was funded by an NIHR School for Primary Care Research Fellowship between October 2015 and September 2017.

**Ethics approval** In GER, UK, PL, CZ, the study has been reviewed and approved by the respective responsible ethics committee, in NL no ethical approval was required. The respective names of all ethics committees as well as reference numbers are as follows: GER: Ethics Committee of the Hamburg Medical Chamber WF-03/17; UK: National Health Service Health Research Authority North East—Newcastle and North Tyneside 2 Research Ethics Committee 17/ NE/0283; PL: Academy of Special Education Ethical Committee 168-2018/2019; CZ: Ethical Committee of the National Monitoring Centre for Drugs and Addiction 180326_EK-NMS.

**Provenance and peer review** Not commissioned; externally peer reviewed.

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
