## [Reviewer comments · BMJ Open]

ARTICLE DETAILS

TITLE (PROVISIONAL)	Understanding Pathways to Stimulant Use: a mixed-methods examination of the individual, social and cultural factors shaping illicit stimulant use across Europe (ATTUNE): study protocol
AUTHORS	Rosenkranz, Moritz; O'Donnell, Amy; Verthein, Uwe; Zurhold, Heike; Addison, Michelle; Liebrechts, Nienke; Rowicka, Magdalena; Barták, Miroslav; Petruželka, Benjamin; Kaner, Eileen; Martens, Marcus-Sebastian

VERSION 1 – REVIEW

REVIEWER	Niloofar Bavarian California State University, Long Beach; United States
REVIEW RETURNED	13-Feb-2019

GENERAL COMMENTS	Thank you for your contribution. Please see my comment below: *Abstract: -The Abstract should make clear what constitutes ATS (this is done in the Introduction, but should also be specifically stated in the Abstract) *Introduction -The authors lump together various forms of ATS, and when discussing things like motive, do not specify which type of ATS matches with the motive they are discussing. -If I am interpreting correctly, the author's are implying a lack of research examining NPS, which is a very inaccurate statement. A handful of systematic reviews have been completed on the NPS literature. Please clarify. -If instead the authors believe European-based studies are limited, they should better articulate why studying this behavior in European countries is warranted, as well as how findings may compare/contrast with studies completed in the U.S. *METHODS -Please define ATTUNE -Since these methods have been used in NPS studies in the US, the authors need to make more clear how they are going to frame each form of ATS they've identified/defined in the Introduction -Rationale for the eligibility criteria should be better articulated -Justification for the study group operationalizations should be provided -A more developed interview guide for the semi-structured interviews should be presented (Module 1). -A more specific survey guide for Module 2 should be presented (e.g., what pre-existing instruments will be used; what is their validity/reliability?) Note: Table 5 is somewhat clear, but the text above is not specific. -The Methods section is the first time the different use groups are
---

	discussed - this should be done in the Introduction
REVIEWER	Ian Hamilton University of York United Kingdom
REVIEW RETURNED	19-Feb-2019
GENERAL COMMENTS	I enjoyed reading about the plans for this study which looks promising and as you say explores a largely under investigated but popular area of drug use. I hope these small suggestions are helpful: In the introduction on page 4 it would be useful to say a little about ATS dependence, how is it defined ? what do we know about incidence & prevalence ? otherwise there is a disconnect between the first and second paragraph. A source supporting the sample size calculation on page 8 would be helpful.

VERSION 1 – AUTHOR RESPONSE

Reviewer: 1	
ABSTRACT	
The Abstract should make clear what constitutes ATS (this is done in the Introduction, but should also be specifically stated in the Abstract)	We have listed the different ATS substances in the abstract now as well
INTRODUCTION	
The authors lump together various forms of ATS, and when discussing things like motive, do not specify which type of ATS matches with the motive they are discussing	A remark and explanation for our approach here was added.
If I am interpreting correctly, the author's are implying a lack of research examining NPS, which is a very inaccurate statement. A handful of systematic reviews have been completed on the NPS literature. Please clarify. If instead the authors believe European-based studies are limited, they should better articulate why studying this behavior in European countries is warranted, as well as how findings may compare/contrast with studies completed in the U.S.	The inaccurate statement was deleted.
METHODS	
Please define ATTUNE	The project title, which the acronym ATTUNE arose out of, was already mentioned in the introduction. To make it more clearly we highlighted the respective letters by formatting them bold.
Since these methods have been used in NPS studies in the US, the authors need to make more clear how they are going to frame each form of ATS they've identified/defined in the Introduction	On our recruitment flyers we name five groups of ATS substances and add examples in colloquial language. Due to our large sample size we expect a sufficient variety of the specific forms of ATS. We have deleted the misleading sentence about NPS.

Rationale for the eligibility criteria should be better articulated	We have revised the respective section and explained the reasons for choosing the eligibility criteria more detailed.
Justification for the study group operationalizations should be provided	We clarified the operationalizations for the study groups in Module 1 as well as Module 2.
A more developed interview guide for the semi-structured interviews should be presented (Module 1).	We provided the interview guidelines as a supplement and mentioned it in the text.
-A more specific survey guide for Module 2 should be presented (e.g., what pre-existing instruments will be used; what is their validity/reliability?) Note: Table 5 is somewhat clear, but the text above is not specific.	We reworked the text passage and pointed out the instruments. We added information about validity and reliability in the table, if available. We desist from attaching the quantitative questionnaire in the supplements because it is very long as a whole (due to many filters resp. parts that only subgroups of respondents will have to answer). Another reason is the already mentioned copyright restrictions of the measures.
The Methods section is the first time the different use groups are discussed - this should be done in the Introduction	We added some information about the targeted user groups in the introduction.
Reviewer: 2	
In the introduction on page 4 it would be useful to say a little about ATS dependence, how is it defined ? what do we know about incidence & prevalence ? otherwise there is a disconnect between the first and second paragraph.	We added some information about ATS dependency in order to better connect first and second paragraph. The definition of ATS dependency does not differ from the definition for dependency of other psychoactive substances, which are widely known: criteria from DSM or ICD. So we desisted to list the criteria here to avoid the introduction becoming too long. In our survey we used the SDS, mentioned this in the text now and listed it in the overview table of measures.
A source supporting the sample size calculation on page 8 would be helpful.	We added a source.

VERSION 2 – REVIEW

REVIEWER	Niloofar Bavarian CSULB, USA
REVIEW RETURNED	16-May-2019
GENERAL COMMENTS	I served as Reviewer #1. I thank the authors for their revision, and attention to reviewer comments. I have just a minor edit request: -It is unclear how anonymity will be maintained (e.g., will consent forms be signed by the research team? will an alias be assigned to each participant?). Please explain. -Please expand on the planned statistical analyses for module 2.

VERSION 2 – AUTHOR RESPONSE

Reviewer: 1	
It is unclear how anonymity will be maintained (e.g., will consent forms be signed by the research team? will an alias be assigned to each participant?). Please explain.	We explained more detailed the procedure regarding informed consent for Module 1 as well as for Module 2.
Please expand on the planned statistical analyses for module 2.	We added some examples of planned statistical analyses. Of course we have many more ideas and plans regarding statistical analyses but in the light of the word limitation of this manuscript we confined ourselves to giving only two more examples.

VERSION 3 – REVIEW

REVIEWER	Niloofar Bavarian CSULB
REVIEW RETURNED	02-Jul-2019
GENERAL COMMENTS	The authors have addressed my lingering issues. I would just ask, with respect to anonymity, for a sentence to be added to explain if an alias will be used for participants in the qualitative interviews.

VERSION 3 – AUTHOR RESPONSE

Dear Reviewer,

thank you for revising the manuscript again.

I understand your request, however, you maybe have missed that we already provided information on how we preserve the anonymity of participants. On page 7 (page 8 in pdf document) of the manuscript we stated:

"A randomly chosen screening ID facilitates the connection of the screening data with the interview whilst maintaining anonymity."

In my opinion, another sentence about this issue would be redundant. Therefore I would like to leave the manuscript as it is.